# Characterization of Engineering Plastics Plasticized Using Supercritical CO_2_

**DOI:** 10.3390/polym12010134

**Published:** 2020-01-06

**Authors:** Masaki Watanabe, Yoshihide Hashimoto, Tsuyoshi Kimura, Akio Kishida

**Affiliations:** Department of Material-Based Medical Engineering, Institute of Biomaterials and Bioengineering, Tokyo Medical and Dental University, Tokyo 101-0062, Japan; masaki.wm.watanabe@jp.ricoh.com (M.W.); hashimoto.atrm@tmd.ac.jp (Y.H.); kimurat.mbme@tmd.ac.jp (T.K.)

**Keywords:** engineering plastics, polycarbonate, polysulfone, polyarylate, supercritical CO_2_

## Abstract

The purpose of this study was to evaluate the physical and chemical properties of engineering plastics processed using supercritical CO_2_. First, we prepared disk-shaped test pieces via a general molding process, which were plasticized using supercritical CO_2_ at temperatures lower than the glass-transition points of engineering plastics. Amorphous polymers were plasticized, and their molecular weight remained nearly unchanged after treatment with supercritical CO_2_. The mechanical strength significantly decreased despite the unchanged molecular weight. The surface roughness and contact angle increased slightly, and electrical properties such as the rate of charging decreased significantly. These results suggest that supercritical CO_2_ could be used for a new molding process performed at lower temperatures than those used in general molding processes, according to the required properties.

## 1. Introduction

Engineering plastics are expected to have various applications because of their desirable mechanical properties, heat resistance, and chemical resistance. Due to their excellent properties, they also attract attention as alternative materials to metals. For example, polycarbonate (PC) has applications in optical devices [1], polysulfone (PSU) is utilized in dialysis membranes [2], and polyarylate (PAR) is used in illuminating devices [3]. However, as these engineering plastics have high mechanical and thermal properties [4], they are difficult to mold into the desired shapes with predictable qualities. As a result, new molding processes such as 3D printing [5,6] and others [7,8] are being studied.

It is well known that supercritical fluids can create unique environments. In particular, supercritical CO_2_ is expected to have applications in various fields, such as pharmaceuticals and semiconductors, because it can be controlled in various forms such as nanoparticles, microcapsules, thin films, and foams [9,10,11,12,13,14,15,16,17,18,19]. There has been extensive research on supercritical fluids using low-molecular-weight compounds and inorganic compounds [20,21]. Recently, research on polymers [16,17,18,19,22,23] has increased considerably because of their potential uses in various fields. For example, McHugh et al. performed systematic studies on the phase behavior of polymers in supercritical fluid solvents [24]. Erdogan has reported that supercritical CO_2_ enables the molding of polymers into fine particles, films, fibers, membranes, composites, and foams [17,19]. In other words, supercritical CO_2_ can plasticize polymers in the low-temperature range below their glass-transition temperature (*Tg*): this plasticization can decrease the viscosity of the polymers such that it is lower than that observed in existing molding processes. Therefore, combining supercritical CO_2_ with molding processes such as injection molding and 3D printing will make molding polymers into various shapes easier than ever. Processing at lower temperatures can also be expected to be similar to general molding processes and may be useful from the viewpoint of green chemistry. However, there have been limited studies reporting the physical properties of engineering plastics plasticized using supercritical fluids. 

Hence, in this study, we subjected various engineering plastics to varying supercritical conditions using supercritical CO_2_ and investigated their physical and chemical properties, such as dynamic strength, surface texture, and wetting, along with their electrical properties.

## 2. Materials and Methods 

### 2.1. Apparatus

Figure 1 shows a schematic of the experiment apparatus. The system mainly consisted of a high-pressure pump (NP-FX-25(J), Nihon Seimitsu Kagaku Co., Ltd., Tokyo, Japan), a supercritical reaction chamber (Koatsu System Co., Ltd., Saitama, Japan), and a back-pressure regulator (26-1700, TESCOM Co., Ltd., Tokyo, Japan). CO_2_ first passed through the preheater (Koatsu System Co., Ltd., Saitama, Japan): it was then further heated by a circulator (MA-4, JULABO JAPAN Co., Ltd., Tokyo, Japan) and pressurized to a supercritical state in the reaction chamber. The sample holder (Koatsu System Co., Ltd., Saitama, Japan) was fixed in the reaction chamber with a processing capacity of 35.2 m^3^/day. The reaction chamber had the following specifications: a design pressure of 29.9 MPa (the regularly used pressure is 25 MPa), a design temperature of 200 °C (the regularly used temperature is 150 °C), and a vessel size of 200 mL.

### 2.2. Materials

Engineering plastics such as poly (l-lactic acid) (PLLA) (4032D, NatureWorks LLC, Minnetonka, USA), polyarylate (PAR) (U-100, Unitika Co., Ltd., Osaka, Japan), polycarbonate (PC) (K-1300Y, Teijin Co., Ltd., Osaka, Japan), PSU (182443, Sigma-Aldrich Co., LLC, Saint Louis, MO, USA), polyphenyl sulfone (PPSU) (428310, Sigma-Aldrich Co., LLC, Saint Louis, MO, USA), polyetherimide (PEI) (700193 Sigma-Aldrich Co., LLC, Saint Louis, MO, USA), polyethyleneterephthalate (PET) (429252, Sigma-Aldrich Co., LLC, Saint Louis, MO, USA), polybutyleneterephthalate (PBT) (190942, Sigma-Aldrich Co., LLC, Saint Louis, MO, USA), and polyamide 6 (PA6) (NL-H01-1011FB, UBE Industries Ltd., Tokyo, Japan) were used as model materials. Table 1 presents the molecular structures, crystal structures, glass-transition points, and melting points of these engineering plastics. These plastics were molded into disks with a diameter of 9 mm and a thickness of 0.2 mm and placed in the sample chamber.

### 2.3. Procedure

The disk-shaped samples were first placed in the sample holder, and then the sample holder was placed in the reaction chamber. CO_2_ (99.9% purity) was passed through a heat exchanger, and this heated CO_2_ was further heated and pressurized to a supercritical state in the reaction chamber. After reaching the desired conditions, the temperature was first lowered to near room temperature, and then the pressure was slowly lowered to obtain an unfoamed sample. To obtain a porous/foamed sample, the pressure was first lowered to atmospheric pressure, and then the temperature was lowered. Plasticization effects due to changes in pressure, temperature, and time were evaluated using polymer samples under the various conditions listed in Table 2. 

### 2.4. Characterization

The appearance of the PLLA pellet after supercritical processing was evaluated visually, and the molecular weight was analyzed using gel permeation chromatography (GPC) (HLC-8320, Tosoh Co., Ltd., Tokyo, Japan) with a column comprised of TSKgel SuperH1000, H3000, H4000, and H5000 (using tetrahydrofuran as the eluent). The tensile strengths of the other plastics before and after supercritical processing were evaluated using an Autograph tensile tester (AGS-X, Shimazu Co., Ltd., Kyoto, Japan) at room temperature. The surfaces were observed via scanning electron microscopy (SEM) (MERLIN, Carl Zeiss Co., Ltd., Oberkochen, Germany) and laser microscopy (Keyence Co., Ltd., Osaka, Japan), and their surface roughness (*Ra*) values were calculated using software. The hydrophilicity of the processed plastics was evaluated using a contact angle meter (FTA1000B, First Ten Angstroms, Inc, Portsmouth, NH, USA). The amount of electric charge was determined using a corona discharge apparatus. 

## 3. Results and Discussion

### 3.1. Supercritical Treatment of PLLA

Figure 2 shows the appearance of the PLLA pellet subjected to supercritical processing by varying the pressure and time. Under a 5-MPa pressure for 1 h (lower pressure), the treatment did not cause changes in appearance, while treatments at 15 MPa and 25 MPa (higher pressure) resulted in both plasticization and melting. During treatment under 15 MPa of pressure for 0.5 h, no change in appearance occurred, but treatment under 15 MPa for 1 and 2 h resulted in melting. A possible explanation for this is that the collision frequency energy of the CO_2_ molecules colliding with the PLLA pellets increased at higher pressures, causing the CO_2_ molecules to disrupt intermolecular relaxation in PLLA. As PLLA pellets retained their shape under low pressures such as 5 MPa and did not do so under pressures greater than 15 MPa, the results suggest that plasticization requires kinetic energy from the CO_2_ molecules. However, the duration of supercritical treatment could also plasticize PLLA under a constant temperature and pressure. That is, the collision frequency and energy of the CO_2_ molecules colliding with PLLA increased with the duration, causing the CO_2_ molecules to disrupt intermolecular relaxation in PLLA. As mentioned above, the PLLA pellets retained their shape during short treatment times such as 0.5 h and failed to do so during treatments with longer durations, such as more than 1 h. This result indicates that plasticization needs to account for the collision frequency of CO_2_ molecules. Therefore, it is essential to consider the effects of temperature, pressure, and time on the molecular mobility of polymers before setting supercritical treatment conditions.

The solubility of CO_2_ in polymers is significantly influenced by pressure rather than temperature [25,26]. In this study, we mostly used a temperature of 150 °C to confirm the effect of pressure. Additionally, in the case of polymers such as PLLA that can be synthesized using organic molecular catalysts [27], the catalyst may be extracted through supercritical processing, with a slight mass loss. Therefore, it is necessary to consider the components of the polymers before performing supercritical treatment. Table 3 compares the molecular weights before and after supercritical treatment. Both the number- and weight-averaged molecular weights of PLLA before and after treatment were almost the same. This means PLLA did not degrade during supercritical treatment. It did not undergo thermal degradation, as it was processed at a temperature 30 °C lower than its melting point.

### 3.2. Plasticization Behavior of Engineering Plastics

Tomasko et al. reported that the *Tg* of several polymers (PMMA, PS, PEMA, PVC, PC, and PET) is lowered by approximately 1 °C for every 1-atm increase in pressure in the 0–92-atm range [16]. They also reported polymer characteristics under supercritical conditions in terms of the solubility of CO_2_ in polymers, the plasticization of polymers, and the rheology of polymer melts with dissolved CO_2_. These findings suggest the possibility of using supercritical CO_2_ in polymer processing. In this study, we tried to determine the plasticization behavior of engineering plastics, which Tomasko et al. did not study.

Figure 3 shows the effects of plasticization in engineering plastics. PAR, PSU, PC, PPSU, and PEI were plasticized. In contrast, PET, PBT, and PA 6 were not plasticized and remained in pellet form. Thus, PAR, PSU PC, PPSU, and PEI can be molded to some extent using supercritical fluids. Amorphous polymers can be plasticized while crystalline polymers cannot be plasticized. In the case of crystalline polymers, the interaction between molecules is strong, and this prevents CO_2_ molecules from breaking the molecular chains of these polymers. In contrast, amorphous polymers have a random morphology, and this can cause CO_2_ molecules to break the molecular chains of these polymers.

Figure 4a–c shows SEM images of a PSU surface. The sample in Figure 4b was formed under nonfoaming conditions and did not have a porous structure; in contrast, the sample in Figure 4c was prepared under foaming conditions and had a random porous structure. This means some plasticized polymers, namely PSU, PAR, PC, PPSU, and PEI, can be processed to obtain a porous structure.

Figure 4d presents the surface roughness (*Ra*) values. The *Ra* values of all samples increased after supercritical treatment, and the rates of *Ra* before and after treatment were 1.46 in PSU, 1.42 in PAR, and 1.87 in PC. 

Supercritical CO_2_ can be used for morphological modifications, and there have been several reports confirming this [16,17,18,19]. High pressure makes a polymer porous. Therefore, a molten polymer with supercritical CO_2_ undergoes foaming during depressurization as CO_2_ escapes. Nishikawa has reported that the solubility of CO_2_ in polymers depends on their molecular structures: specifically, polymers containing carbonyl groups can easily absorb CO_2_ [28]. Hence, in this study, the reason why PC and PAR showed higher *Ra* values than PSU did could be the amount of dissolved CO_2_.

Supercritical treatment roughens polymer surfaces, and such polymers are useful in several applications. For example, porous polymer particles such as particles from gas-saturated solutions can be used in cosmetics and medical applications. In medical applications, roughness is one of the important factors influencing cell adhesion and protein adsorption, so there is a possibility of using these polymers in medical and dental prosthesis. However, these polymers cannot be used in optical applications, where transparency is important.

### 3.3. Physical Properties

We evaluated the surface roughness (*Ra*) and contact angles of PAR, PSU, and PC before and after supercritical treatment. As PAR, PSU, and PC have similar molecular compositions comprising carbonyl groups and bisphenol A, we used them as representative samples. The treatment conditions are given in Table 2, and these samples did not have porosity.

Figure 5 shows the contact angles of the samples. The contact angles of all samples increased after supercritical treatment, and the rates of contact angles before and after treatment were 1.15 in PSU, 1.32 in PAR, and 1.36 in PC. Increases in roughness caused increases in the contact angles.

Next, we evaluated the tensile strength of PSU. For measuring the tensile strength, we used three test pieces. One was an unprocessed test piece, the second was a porous test piece formed via supercritical treatment, and the third was a nonporous test piece also formed through supercritical treatment. Figure 6 shows the tensile behavior. Supercritical treatment caused the Young’s modulus and maximum yield stress to be reduced, and the elongation also lowered. In addition, the porous structure also decreased the physical properties. This means supercritical treatment reduces the strength of plasticized materials such as PSU.

### 3.4. Electrical Properties

We evaluated the potential decay behavior and rate of charging (*V*) for PAR, PSU, and PC. The corona discharge was done for 2 s, and the measurement was done at 10 s after and 60 s after discharge. Figure 7 presents the potential decay behavior of these materials. The initial amount of charge in the plasticized samples was lower than that in the unprocessed samples. The potential decay in the unprocessed samples was extremely rapid, and the rate of charging was 1.43 in PSU, 1.36 in PAR, and 1.79 in PC. In contrast, the plasticized samples showed a slow potential decay, and the rate of charging was 1.11 in PSU, 1.05 in PAR, and 1.05 in PC. This implies that the plasticized samples do not easily charge and discharge. According to the temperature-programmed desorption–mass spectrometry (TPD–MS) results (Appendix A), CO_2_ was continuously released even after the samples were maintained at temperatures of 80 °C for 2 h, and the rate of release tended to gradually decrease. Therefore, the polymer plasticized by the supercritical CO_2_ also included residual CO_2_. The plasticized polymers had a smaller potential decay and rate of charging than the unplasticized ones. As the CO_2_ present in the polymer acted as an insulator, the transfer of electric charges between the atmosphere and polymers was suppressed, thus yielding a reduced rate of charging. The rate of charging in PC and PAR tended to be smaller than that in PSU. PC and PAR have a carbonyl group that is similar to CO_2_ in terms of its molecular structure. Hence, we inferred that the rate of charging was reduced by the carbonyl group positively holding CO_2_.

To verify the hypothesis stated above, we evaluated the potential decay behavior when CO_2_ was removed from the samples through high-temperature vacuum drying. Figure 7 shows the change in the rate of charging after drying. The rate of charging in the dried samples tended to be higher than in the undried samples. Some of the CO_2_ may have been removed during vacuum drying. That is, the removal of CO_2_ facilitated a transfer charge to the atmosphere because the potential decay behavior of the dried samples was close to what it was before supercritical treatment. Furthermore, changes in the rate of charging in PC and PAR were significant. These polymers have a carbonyl group that retains large amounts of CO_2_; hence, these polymers could have contained significant amounts of CO_2_ that could be removed through vacuum drying.

To explain the above results, it is necessary to consider the solubility of CO_2_ in polymers. According to Masuoka et al. [25,26], the solubility of CO_2_ in polymers increases almost linearly with increasing pressure and decreases with increasing temperature. Furthermore, the diffusion coefficient of CO_2_ in polymers increases with increasing solubility of the high-pressure gas at low temperatures. This indicates that dissolution of the high-pressure gas promotes plasticization. Tomasko et al. reported that pressure and temperature gradients can drive bubble nucleation from dissolved CO_2_ because pressure drops or temperature increases make solutions supersaturated. Thus, the potential decay in the polymers was suppressed by supercritical treatment because insulating CO_2_ was dissolved in the polymer and bubbles were formed during depressurization. Moreover, it is also possible that the remaining CO_2_ suppressed charge leakage. However, if all the remaining CO_2_ in the polymers were converted into bubbles, it is conceivable that charges would leak through a region where bubbles do not exist. Further investigation is necessary to determine if any remaining CO_2_ is converted into bubbles or remains between the polymer molecules as a plasticizer.

## 4. Conclusions

Supercritical CO_2_ can plasticize amorphous engineering plastics at temperatures lower than their glass-transition point: it can control hydrophilicity and electrical properties without substantial changes in molecular weight. While the mechanical strength was altered, supercritical CO_2_ could be used for a new molding process at temperatures lower than those used in general molding processes. In this study, we selected specific temperatures and pressures for our experiments. We analyzed changes in the physical properties of the selected polymers plasticized using supercritical CO_2_. However, to develop this supercritical treatment process as a new molding method, it is necessary to perform experiments under other temperatures and pressures.

## Figures and Tables

**Figure 1 polymers-12-00134-f001:**
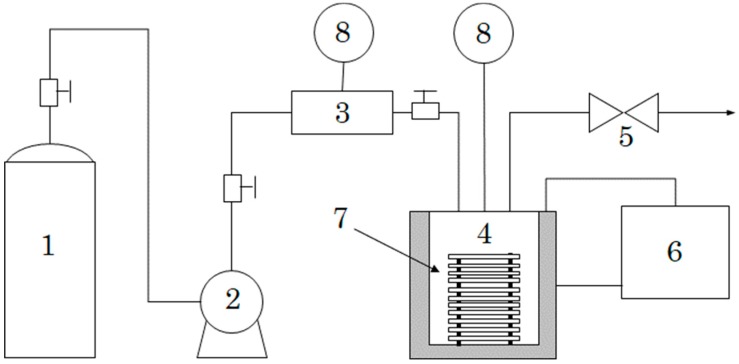
Schematic representation of the newly devised experimental apparatus: (1) CO_2_ tank, (2) CO_2_ pump, (3) preheater, (4) reaction chamber, (5) back-pressure regulator, (6) circulator, and (7) sample holder. (8) thermometer.

**Figure 2 polymers-12-00134-f002:**
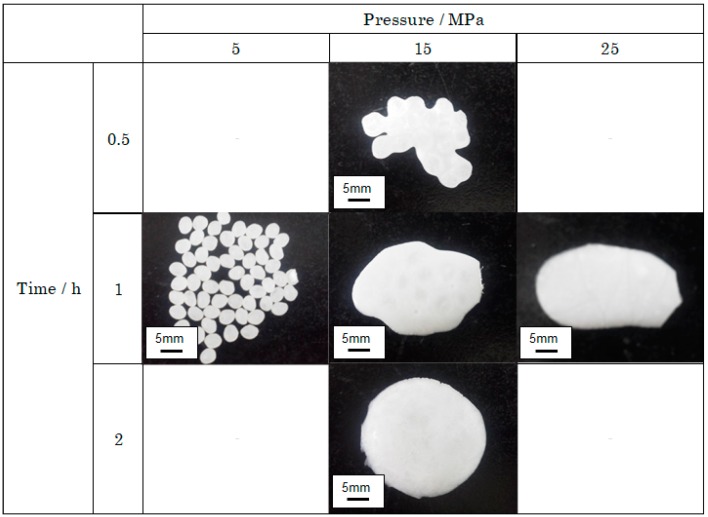
Appearance of PLLA pellets after supercritical treatment.

**Figure 3 polymers-12-00134-f003:**
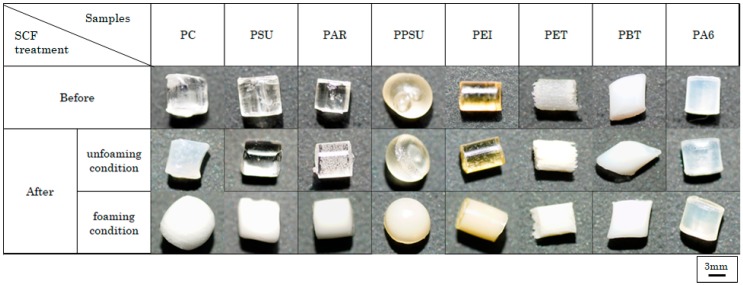
Appearance of pellets formed using different engineering plastics before and after supercritical treatment.

**Figure 4 polymers-12-00134-f004:**
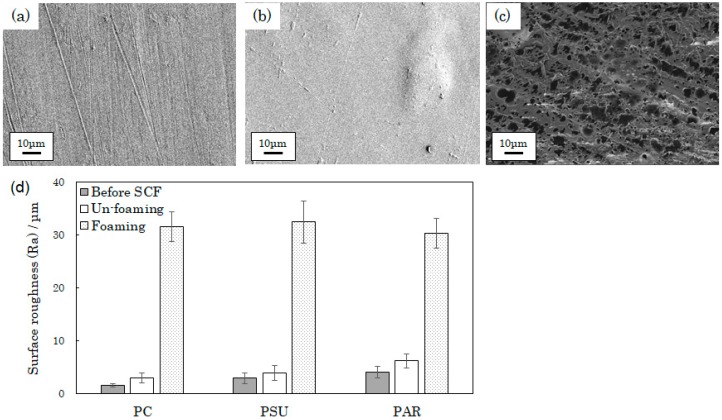
Surface properties of engineering plastics treated with supercritical CO_2_: SEM images of PSU (**a**) before supercritical treatment, (**b**) after supercritical treatment under nonfoaming conditions, and (**c**) after supercritical treatment with foaming. (**d**) Surface roughness (*Ra*) values of PC, PSU, and PAR before and after supercritical treatment (with and without foaming).

**Figure 5 polymers-12-00134-f005:**
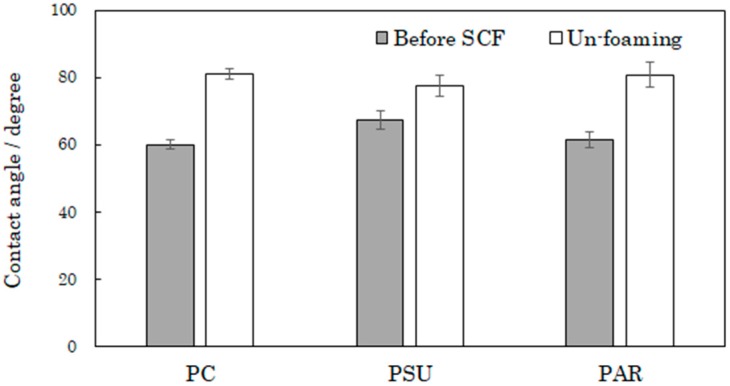
Contact angles of PC, PSU, and PAR before and after supercritical treatment.

**Figure 6 polymers-12-00134-f006:**
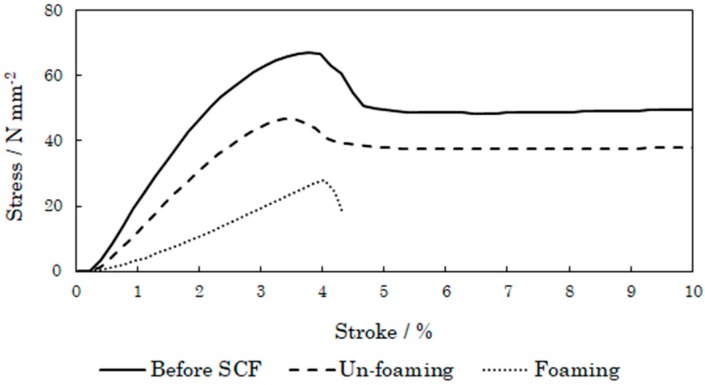
Tensile strengths of PSU before and after supercritical treatment.

**Figure 7 polymers-12-00134-f007:**
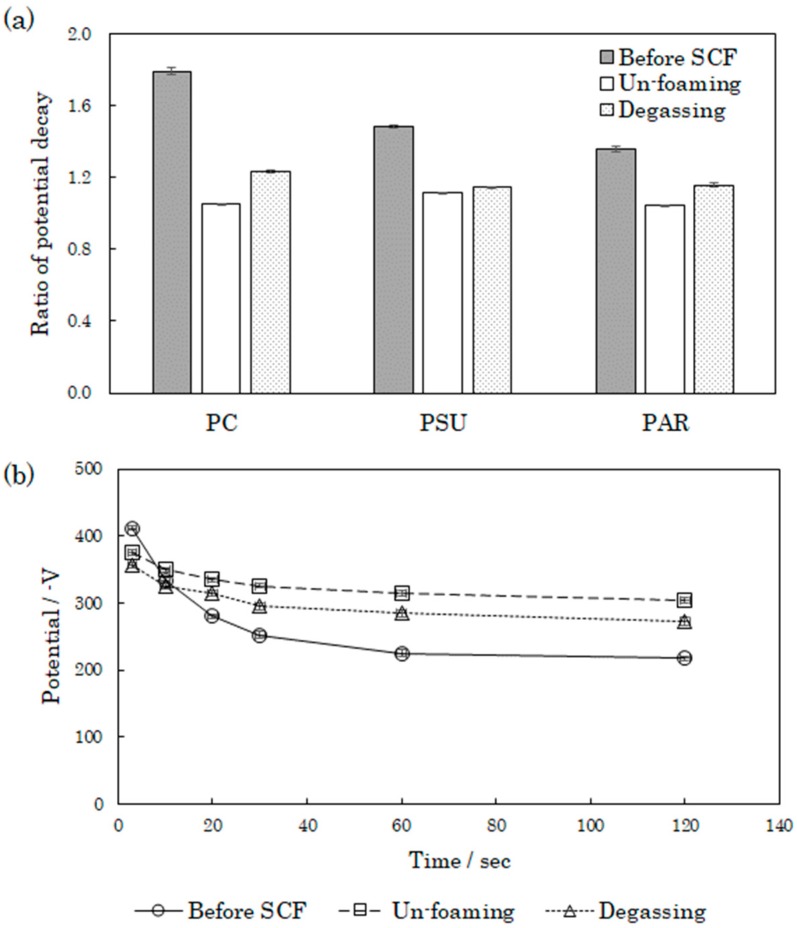
Electrical properties of different engineering plastics: (**a**) potential decay and (**b**) potential decay curve.

**Table 1 polymers-12-00134-t001:** Molecular structures, crystal structures, glass-transition points (*T_g_*), and melting points (*T_m_*) of engineering plastics.

Polymer (Abbreviation)	Molecular Structure	Crystal Structure	*T_g_*/°C	*T_m_*/°C
Polycarbonate (PC)	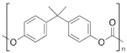	Amorphous	150	-
Polysulfone (PSU)	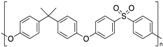	Amorphous	190	-
Polyarylate (PAR)	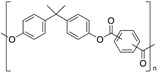	Amorphous	193	-
Polyetherimide (PEI)	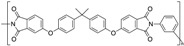	Amorphous	217	-
Polyphenylsulfone (PPSU)	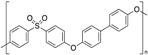	Amorphous	220	-
Polylactic acid (PLLA)	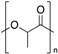	Crystalline	60	175
Polyethylene terephthalate (PET)	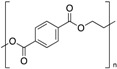	Crystalline	69	260
Polybutylene terephthalate (PBT)	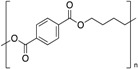	Crystalline	50	225
Polyamide 6 (PA 6)	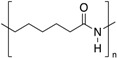	Crystalline	50	225

**Table 2 polymers-12-00134-t002:** Experimental conditions.

Number	Polymer	Temperature/°C	Pressure/MPa	Time/h
1	PLLA	150	5	1
2			15	0.5
3			15	1
4			15	2
5			25	1
6	PC	120	25	1
7	PSU	150	25	1
8	PAR	150	25	1
9	PEI	150	25	1
10	PPSU	150	25	1
11	PET	150	25	1
12	PBT	150	25	1
13	PA6	150	25	1

**Table 3 polymers-12-00134-t003:** Molecular weights of PLLA before and after supercritical treatment.

	Weight-Averaged Molecular Weight (*Mw*)	Number-Averaged Molecular Weight (*Mn*)
Before	186,000 ± 1000	138,000 ± 1000
After	182,000 ± 1000	131,000 ± 1000

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
