# Peer review of "Characterization of Engineering Plastics Plasticized Using Supercritical CO2"

_polymers, 2020, doi:10.3390/polym12010134_

Round 1

Reviewer 1 Report

REVIEW POLYMERS 664425

This work shows the impact on various parameters of several polymers of interest in moulding applications, which causes supercritical CO2 treatment. Although the results are interesting per se, a deeper discussion of why they are in relation to the application is lacking. In addition, a broader study of conditions is also missing. Conclusions are drawn from the exploration under too narrow pressure and temperature conditions. In fact, as far as temperature is concerned, only one is explored. In exchange, the results are clearly presented, and figures 2 and 3 are very explicit, beautiful and illustrative. The document in general is quite well written. It could be improved by removing abbreviations that are not appropriate in formal written language and putting all the results in simple past tense. In any case, a review by English-speaking editors would be advisable.

Specifically, there is no better justification in the introduction section for the need for this research in the field of the use of CO2 in the processing of these polymers. Particularly relevant is the selection of test conditions. Are they relevant to moulding processes where CO2 would be used as a molding agent? Other? In particular, the authors should justify the pressure and above all, the chosen temperature (150 ºC) in that it is very high for the general applications of supercritical CO2. In the case of crystalline polymers, 150°C is below the melting temperature, but it is very close to the Tg of most amorphous polymers. It is well known, and should have been cited, the work of (Tomasko et al., 2003) in which it is shown that the Tg of several polymers (PMMA, PS, PEMA, PVC, PC, PPO and PET) is lowered by approximately 1 ºC for every 1 atm in the interval of 0 to 92 atm. This means that at 250 bar the Tg would be lowered very much. This fact could explain most of the observations in this paper in relation to the amorphous polymers. In this sense, the final conclusion on the impact of CO2 on the plasticising effect at temperatures lower than Tg should be eliminated. It is previously shown and justified by other authors.

Although it is also well known, for readers unfamiliar with scCO2 technology, the plasticizing effect of scCO2 on polymers should be remembered in relation to its high solubility and retention. This last effect seems to be important to explain the electrical properties of the polymers (PAR, PSU and PC). There are several reviews that can be cited to explain these issues (among others):

Tomasko, D. L., Li, H., Liu, D., Han, X., Wingert, M. J., Lee, L. J., & Koelling, K. W. (2003). A Review of CO2 Applications in the Processing of Polymers. Industrial and Engineering Chemistry Research, 42(25), 6431–6456. https://doi.org/10.1021/ie030199z

Kiran, E. (2009). Polymer miscibility, phase separation, morphological modifications and polymorphic transformations in dense fluids. Journal of Supercritical Fluids, 47(3), 466–483. https://doi.org/10.1016/j.supflu.2008.11.010

Nalawade, S. P., Picchioni, F., & Janssen, L. P. B. M. (2006). Supercritical carbon dioxide as a green solvent for processing polymer melts: Processing aspects and applications. Progress in Polymer Science (Oxford), 31(1), 19–43. https://doi.org/10.1016/j.progpolymsci.2005.08.002

Kiran, E. (2016). Supercritical fluids and polymers - The year in review - 2014. Journal of Supercritical Fluids, 110, 126–153. https://doi.org/10.1016/j.supflu.2015.11.011

With regard to crystalline polymers, obviously if they have been operated well below the melting temperature, there has been hardly any modification during processing with CO2. However, in the case of PLLA there is a slight loss of mass that could correspond to the extraction of some components. Please comment.

In relation to the final application of these polymers there is also a little more explanation of the expected properties in the moulds. It would be convenient to explain in this sense the electrical measurements. Why are they relevant? what would be the ideal?

Contrary to what the authors summarise, it appears that contact with CO2 alters all the properties of the polymers quite significantly, so it is not clear whether it is really an ideal solvent for use in moulding. A more realistic analysis should be made, probably in relation to each polymer and according to the property.

In summary, I recommend a more extensive introduction to the topic both to establish the state of the art and to further frame the research interest in the concrete application of moulding. In addition it is essential to introduce in the discussion of the results the important effect of the depression of Tg by the CO2 in amorphous polymers.

Author Response

Thank you very much for reviewing our manuscript and providing valuable comments. We have carefully read the comments and suggestions and have revised the manuscript accordingly. We have provided our responses to your comments and suggestions below.

Comment 1

This work shows the impact on various parameters of several polymers of interest in moulding applications, which causes supercritical CO2 treatment. Although the results are interesting per se, a deeper discussion of why they are in relation to the application is lacking. In addition, a broader study of conditions is also missing. Conclusions are drawn from the exploration under too narrow pressure and temperature conditions. In fact, as far as temperature is concerned, only one is explored. In exchange, the results are clearly presented, and figures 2 and 3 are very explicit, beautiful and illustrative. The document in general is quite well written. It could be improved by removing abbreviations that are not appropriate in formal written language and putting all the results in simple past tense. In any case, a review by English-speaking editors would be advisable.

Answer 1

In this study, we mainly examined the characteristics of the polymers under only one temperature. This is because the solubility of CO2 in the polymers is based on the following studies stating that pressure has more influence on the solubility than temperature.

Y.Sato, M.Yurugi, K.Fujiwara, S.Takishima, H.Masuoka, “Solubilities of carbon dioxide and nitrogen in polystyrene under high temperature and pressure”, Fluid Phase Equilibria, 125(1-2), 129-138 (1996). Y.Sato, T.Takikawa, S.Takishima, H.Masuoka, “Solubilities and diffusion coefficients of carbon dioxide in poly(vinyl acetate) and polystyrene”, J.Supercritical Fluids, 19(2), 187-198 (2001).

In this study, the effect of pressure was confirmed using PLLA, and engineering plastics were plasticized under specific temperature and pressure conditions to analyze the changes in their physical properties. In future, we will conduct studies under other temperature and pressure conditions and report the same. We removed those abbreviations that were not appropriate, and revised the results so that they are in the past tense.

Comment 2

Specifically, there is no better justification in the introduction section for the need for this research in the field of the use of CO2 in the processing of these polymers. Particularly relevant is the selection of test conditions. Are they relevant to moulding processes where CO2 would be used as a molding agent? Other? In particular, the authors should justify the pressure and above all, the chosen temperature (150 ºC) in that it is very high for the general applications of supercritical CO2. In the case of crystalline polymers, 150°C is below the melting temperature, but it is very close to the Tg of most amorphous polymers. It is well known, and should have been cited, the work of (Tomasko et al., 2003) in which it is shown that the Tg of several polymers (PMMA, PS, PEMA, PVC, PC, PPO and PET) is lowered by approximately 1 ºC for every 1 atm in the interval of 0 to 92 atm. This means that at 250 bar the Tg would be lowered very much. This fact could explain most of the observations in this paper in relation to the amorphous polymers. In this sense, the final conclusion on the impact of CO2 on the plasticising effect at temperatures lower than Tg should be eliminated. It is previously shown and justified by other authors.

Answer 2

The temperatures were all set at 30 °C lower than the Tg of the amorphous polymers, namely 120 °C for PC, and 150 °C for the others. Tomasko's study is cited in the revised manuscript; however, as there is no discussion regarding the physical properties of the plasticized polymers, we considered this study to be a novel one, examining the physical properties after plasticization. According to your suggestion, in the discussion on plasticization in the revised manuscript, we have cited Tomasko's report. This research focuses on using CO2 as a plasticizer for engineering polymers.

Comment 3

Although it is also well known, for readers unfamiliar with scCO2 technology, the plasticizing effect of scCO2 on polymers should be remembered in relation to its high solubility and retention. This last effect seems to be important to explain the electrical properties of the polymers (PAR, PSU and PC). There are several reviews that can be cited to explain these issues (among others):

Answer 3

We have included citations and explanations regarding the electrical characteristics. As none of the four papers cited discusses electrical characteristics, we considered our study to be unique as it examined the electrical characteristics of the plasticized polymers.

Comment 4

With regard to crystalline polymers, obviously if they have been operated well below the melting temperature, there has been hardly any modification during processing with CO2. However, in the case of PLLA there is a slight loss of mass that could correspond to the extraction of some components. Please comment.

Answer 4

We have mentioned that there is a possibility of mass loss depending on the synthesis method, which is not limited to PLLA.

Reviewer comment 5

In relation to the final application of these polymers there is also a little more explanation of the expected properties in the moulds. It would be convenient to explain in this sense the electrical measurements. Why are they relevant? what would be the ideal?

Answer

We aim to use the scCO2-treated polymers as biomaterials in many possible applications. For biomaterials, the hydrophilicity / hydrophobicity of the material surface, surface roughness, surface potential, and hardness influence protein adsorption and cell adhesion. In this study, the electrical property evaluation was conducted considering possibilities of utilizing these treated polymers as biomaterials because protein adsorption is affected by the electrical properties. If these supercritically treated polymers show clear differences in their electrical properties, then they can significantly influence cell adhesion and protein adsorption

Reviewer comment 6

Contrary to what the authors summarise, it appears that contact with CO2 alters all the properties of the polymers quite significantly, so it is not clear whether it is really an ideal solvent for use in moulding. A more realistic analysis should be made, probably in relation to each polymer and according to the property.

Answer

We should determine whether CO2 is the ideal plasticizer for molding different types of polymers. This study is a first step in this direction. In future, we intend to conduct a realistic analysis according to the properties of different polymers.

Reviewer comment 7

In summary, I recommend a more extensive introduction to the topic both to establish the state of the art and to further frame the research interest in the concrete application of moulding. In addition it is essential to introduce in the discussion of the results the important effect of the depression of Tg by the CO2 in amorphous polymers.

Answer

To expand the range of the study topic for covering molding applications, we have mentioned that there is a possibility that molding processes using supercritical fluids can be applied to produce fine particles, coatings, and foams.

Reviewer 2 Report

This manuscript presents evaluate the physical and chemical properties of engineering plastic processed using supercritical carbon dioxide.

In this work, it is presented physical and chemical properties, such as dynamic strength, surface texture, and wetting and electric properties, of different commercial engineering plastics processed using supercritical carbon dioxide.

All the conclusions are supported by the data and suggest that supercritical carbon dioxide could be used for a new mold process at lower temperatures than in general mold process, according to the required characteristics.

I find this work interesting and the quality of writing is very good. I believe this article should be published in Polymers.

Author Response

Thank you for your encouraging comments.

Round 2

Reviewer 1 Report

My general impression of this second revision is that the authors have made  too brief a response to what I was asking them. They have introduced the references I suggested, more comments on the applications of CO2 -moulded polymers and some more explanation on some outstanding issues regarding the results obtained. But what is still missing is a broader spectrum of operating conditions and a greater depth of analysis of the results in relation to the final application of the molding of the tested polymers by the high pressure CO2.  In any case, the data provided are of interest per se and the work has sufficient scientific content and editorial quality for publication.   

I suggest is to eliminate both the sentence in line 139 and the last paragraph of conclusions in which the authors point out their future work. It is not relevant.

Move this paragraph "The solubility of CO2 in polymers is significantly influenced by pressure rather than temperature  [25,26]. Additionally, in case of polymers such as PLLA that can be synthesized using organic molecular catalysts [27], the catalyst may be extracted through supercritical processing, with a slight mass loss. Therefore, it is necessary to consider the components of the polymers before performing supercritical treatment.” to line 130 before “ Table 3 compares the molecular weights before and after supercritical treatment". It is a more adequate location since you provide the framework for the next work.

This sentence is repited in the introduction and R & D. Choose the best place to leave it. "Tomasko et al. reported the possibility of utilizing CO2 in polymer processing in terms  of the solubility of CO2 in polymers, plasticization of polymers, and rheology of polymer melts with dissolved CO2"